# Predicting Sensitivity to Adverse Lifestyle Risk Factors for Cardiometabolic Morbidity and Mortality

**DOI:** 10.3390/nu14153171

**Published:** 2022-08-01

**Authors:** Hugo Pomares-Millan, Alaitz Poveda, Naemieh Atabaki-Pasdar, Ingegerd Johansson, Jonas Björk, Mattias Ohlsson, Giuseppe N. Giordano, Paul W. Franks

**Affiliations:** 1Department of Clinical Sciences Malmö, Lund University Diabetes Centre, Lund University, 21428 Malmö, Sweden; hugo.pomares-millan@med.lu.se (H.P.-M.); alaitz.poveda@med.lu.se (A.P.); naeimeh.atabaki_pasdar@med.lu.se (N.A.-P.); giuseppe.giordano@med.lu.se (G.N.G.); 2Department of Public Health and Clinical Medicine, Umeå University, 90187 Umeå, Sweden; ingegerd.johansson@umu.se; 3Division of Occupational and Environmental Medicine, Lund University, 22363 Lund, Sweden; jonas.bjork@med.lu.se; 4Clinical Studies Sweden, Forum South, Skåne University Hospital, 22185 Lund, Sweden; 5Computational Biology and Biological Physics Unit, Department of Astronomy and Theoretical Physics, Lund University, 22100 Lund, Sweden; mattias.ohlsson@thep.lu.se; 6Center for Applied Intelligent Systems Research, Halmstad University, 30118 Halmstad, Sweden; 7Department of Nutrition, Harvard T.H. Chan School of Public Health, Boston, MA 02115, USA

**Keywords:** cardiometabolic risk factors, risk assessment, quantile random forests, prediction interval, sensitivity, lifestyle

## Abstract

People appear to vary in their susceptibility to lifestyle risk factors for cardiometabolic disease; determining *a priori* who is most sensitive may help optimize the timing, design, and delivery of preventative interventions. We aimed to ascertain a person’s degree of resilience or sensitivity to adverse lifestyle exposures and determine whether these classifications help predict cardiometabolic disease later in life; we pooled data from two population-based Swedish prospective cohort studies (n = 53,507), and we contrasted an individual’s cardiometabolic biomarker profile with the profile predicted for them given their lifestyle exposure characteristics using a quantile random forest approach. People who were classed as ‘sensitive’ to hypertension- and dyslipidemia-related lifestyle exposures were at higher risk of developing cardiovascular disease (CVD, hazards ratio 1.6 (95% CI: 1.3, 1.91)), compared with the general population. No differences were observed for type 2 diabetes (T2D) risk. Here, we report a novel approach to identify individuals who are especially sensitive to adverse lifestyle exposures and who are at higher risk of subsequent cardiovascular events. Early preventive interventions may be needed in this subgroup.

## 1. Introduction

There is growing recognition that people vary in their susceptibility to environmental risk factors for cardiometabolic diseases, suggesting that one-size-fits-all public health recommendations are unlikely to yield optimal results. Early identification of individuals who are most likely to develop diseases like type 2 diabetes (T2D) and cardiovascular disease (CVD) is desirable, as efficacious therapies (both lifestyle and pharmacologic) exist that can help prevent these diseases [1]. Moreover, once manifest, T2D and CVD often cause life-threatening health complications that are often difficult and costly to treat [2].

Most statistical models examining susceptibility to lifestyle risk factors, from which public health recommendations are drawn, assume that a given lifestyle exposure conveys a similar effect on disease risk throughout the target population, with variability in these effects either viewed as a consequence of measurement error [3] or ignored. However, some of this variability is likely to reflect between-person differences in the effects of unhealthful lifestyle exposures, with some people more susceptible to the adverse effects of these exposures than others.

Predictive modeling often provides a point estimate that represents a response to be anticipated for; yet, in precision medicine a range of values where an effect would be expected to fall may prove more informative for the design of preventive measures rather than a single estimate. Thus, prediction intervals (PIs) allow examining a future series of values for each individual with a given probability, making them potentially useful for identifying where the future value is likely to appear.

Identifying subpopulations who are especially sensitive to adverse lifestyle exposures may help optimize the delivery of cardiometabolic disease prevention programs, especially when resources are lacking [1,4]. In aging and diseased individuals, conditions such as frailty syndrome and nutritional deficiencies often coexist with cardiometabolic disease (i.e., T2D and hypertension) [5]; however, it remains unclear whether vulnerability status associated with adverse environments can be present in disease-free individuals. Here, we used a machine learning approach [6] to differentiate error from true between-individual variability in susceptibility to lifestyle risk factors for T2D and CVD. Accordingly, we identified the subgroup of sensitive individuals and assessed the degree to which this classification aids the prediction of incident disease and premature mortality.

## 2. Materials and Methods

### 2.1. Study Design and Participants

The Västerbotten Health Survey (Västerbottens hälsoundersökning; VHU) [7,8] is a prospective, population-based cohort study designed to monitor and improve health of the general population in Västerbotten county, northern Sweden. Adults residing in Västerbotten are invited to attend their primary care center to undertake a baseline clinical examination and complete detailed lifestyle questionnaires during the calendar years of their 40th, 50th, and 60th birthdays. We used data derived from VHU (n = 42,887) in our analyses. A total of 7039 of these participants were born outside Sweden, and the current analysis focused only on the Swedish-born contingent of VHU. Participants in whom diabetes or cardiovascular disease were diagnosed at baseline (n = 408) were also removed to minimize biases that can occur when people with disease diagnoses are asked to self-report their lifestyle behaviors. Participants with two health examinations between 1985 through 2016 (with ~10 years between each visit) were included in the final dataset, which comprised 35,440 participants.

### 2.2. MDCS

The Malmö Diet and Cancer Study (MDCS) is a prospective, population-based cohort study conducted between 1991 and 1996. All men and women residing in the city of Malmö, southern Sweden born between 1923 to 1950 were invited to participate. Up to 30,446 participants (~40% men) completed the baseline assessment [9,10,11]. Glycemic and lipid traits were assessed in a subset of participants, the *MDCS Cardiovascular Cohort* (MDCS-CC; n = 6103), who were randomly selected for assessment of cardiometabolic risk markers between 1991 and 1994 [12]. As with the VHU cohort, data from non-Swedish participants and those with prevalent diabetes or CVD were removed prior to analysis. In total, a maximum of 18,067 CC participants were included in the analysis from MDCS or MDCS-CC (see flowchart in Figure 1).

### 2.3. Cardiometabolic Risk Markers

Clinical assessment methods in VHU [7] and MDCS are reported elsewhere [9,12]. Briefly, height and weight were measured with calibrated stadiometer and weighing scales respectively, with participants wearing light clothing and no shoes. Body mass index (BMI) was calculated as the body weight in kilograms divided by height in meters squared. Systolic and diastolic blood pressures were measured with participants resting supine, using either manual or automated sphygmomanometers. Peripheral blood was drawn after overnight fasting, and a venous blood sample was drawn two hours after the administration of a 75 g oral glucose load (the latter only in VHU). Blood glucose (i.e., fasting and 2 h glucose), total cholesterol, and triglyceride levels were then measured using a Reflotron bench-top analyzer (Roche Diagnostics Scandinavia AB); HbA1c was measured only in MDCS-CC using standard procedures at the Department of Clinical Chemistry, University Hospital Malmö. High-density lipoprotein cholesterol (HDL-C) was also measured, and low-density lipoprotein cholesterol (LDL-C) was calculated using the Friedewald formula [13]. In September 2009, blood lipids and blood pressure measurements in VHU changed; thereafter, blood pressure was measured twice in a sitting position and averaged. Triglycerides and total cholesterol levels were analyzed using standardized chemical analysis in the hospital clinical biochemistry laboratory. Validated conversion equations were used to adjust the blood pressure and lipids measurements taken before and after September 2009 [14]. For participants on lipid-lowering and/or blood pressure lowering medications, lipid levels and/or blood pressure levels were corrected by adding published constants (+0.208 mmol/L for triglycerides, +1.347 mmol/L for total cholesterol, −0.060 mmol/L for HDL-C, +1.290 mmol/L for LDL-C, +15 mm Hg for systolic, and +10 mm Hg for diastolic blood pressure) suggested in the literature [15,16]. Cardiometabolic trait values outside the thresholds for plausible values suggested by VHU data managers were considered outliers and removed in all datasets (Appendix A).

### 2.4. Lifestyle and Dietary Assessments

For both Swedish cohorts, all participants were requested to complete a self-administered, validated, comprehensive lifestyle questionnaire during each visit, which queried socioeconomic factors, physical/mental health, quality of life, social network and support, working conditions, and alcohol/tobacco use. In VHU, physical activity was assessed using the modified version of the International Physical Activity Questionnaire [17,18], and a validated semiquantitative food frequency questionnaire (FFQ), designed to capture habitual diet over the last year, was used to obtain information on various dietary factors [19]. In 1996, the FFQ was reduced from 84 to 66 items by merging similar items and removing those deemed redundant. For MDCS, a modified diet history method consisting of a 7-day food diary covering all cooked meals and a 168-item FFQ covering the noncooked meals for the previous year were administered. Moreover, a 1 h interview was used to determine portion sizes, cooking methods and food choices. Nutrient and energy contents were calculated using the Swedish Food Composition Database (https://www.livsmedelsverket.se/en/food-and-content/naringsamnen/livsmedelsdatabasen; accessed on 16 February 2021), which is based on meal frequency and portion size. In VHU, food intake level (FIL) was calculated as total energy intake (TEI) divided by estimated basal metabolic rate; individuals with extreme TEI (below the fifth and above the 97.5th percentile of food intake level) were excluded from the analyses [20]. Observations with lifestyle values considered biologically implausible were removed (Appendix A). Written, informed consent was obtained from all living participants at enrolment into VHU and MDCS. VHU study was approved by the Region Ethical Review Board in Umeå and MDCS by the Ethical Committee at Lund University (LU 51-90).

### 2.5. Outcome Ascertainment

Data pertaining to medical diagnoses and mortality were retrieved through record linkage from the National Board of Health and Welfare in Sweden until 31 December 2019. Using each participant’s unique personal identification number, the following diagnosis codes were retrieved: ICD-9 code 250 and ICD-10 codes E11.0–E11.9 for T2D; for the composite CVD outcome, ICD-9 code 410 and ICD-10 code I21 were used for myocardial infarction (MI), and ICD-9 codes 430, 431, and 433–436 and ICD-10 codes I60, I61, I63 and I64 for stroke. The first date of a registered event was selected as the outcome for the current analyses.

### 2.6. Statistical Analysis

All numeric predictors were inverse-normalized to correct skewness, and the derived ordinal variables were treated as continuous variables in subsequent analyses. From an environment-wide association study (EWAS) described elsewhere [21], we prioritized (~300) environmental risk factors that were statistically significant at the corrected *p*-value threshold after multiple testing. We retrieved 167 predictors for BMI, 49 for systolic blood pressure, 47 for diastolic blood pressure, 87 for total cholesterol, 108 for triglycerides, 50 for HDL-C, 21 for LDL-C, 43 for fasting glucose, and 58 for 2 h glucose [22]. Categorical exposure variables with more than two levels were dichotomized into dummy variables. Nutrient data were adjusted for TEI with the residual method [23] to minimize confounding by energy intake and basal energy requirement. We removed correlated (>80%) and zero-variance predictors to minimize the multiple testing burden [24] (Appendix A). For all datasets, we assumed missingness at random [25], and environmental predictor variables with <50% missingness were imputed with the missForest package from R software using a nonparametric approach for mixed data type, to allow a complete case analysis suitable for the random forest algorithm; continuous predictor variables were verified by the mean squared error (MSE) and categorical predictors were verified by the proportion falsely classified (PFC) [26].

We randomly partitioned each dataset into training (50%) and testing (50%) sets to ensure a sufficient number of events per category for the time-to-event analysis in the testing set. The training set was used to fit quantile regression forest (QRF) models for predictors associated with the cardiometabolic traits, and the testing set was used to predict future intervals. Multicollinearity of the variables within these models was assessed using the variance inflation factor, with variables with values > 10 removed [27]. All models were adjusted for age, age^2^, sex, FFQ version, BMI (when not as response variable), follow-up time, and fasting status (for glycemic and lipid models). We utilized QRFs [6], an extension of the supervised machine learning technique *random forest*, which is an ensemble of simultaneous decision trees derived from bootstrapped samples [28]. Furthermore, we set PIs at 90% probability (fifth and 95th quantiles) to minimize false positives ((1 − *α*) × 100%). The PIs were constructed from the conditional quantiles of the trait response predicted by QRFs. Briefly, the prediction intervals of a trait response *Y* given the environmental predictors *X* was built by *I*(*x*) = [*q α*/2 (*Y*|*X* = *x*), *q* 1 − *α*/2 (*Y*|*X* = *x*)]. Thus, the 90% prediction interval for the trait value was estimated using Equation (1).
(1)I(x)=[q 0.05 (Y|X=x), q 0.95 (Y|X=x)],
where, for a given *x*, the trait response lies within the interval *I*(*x*) with high probability. For VHU, on the basis of the obtained PIs per trait, we defined two groups of persistence: those above the 90% PI (‘sensitive’) and below 90% PI (‘resilient’). However, in MDCS, it was not possible to consider two consecutive measures. Instead, QRFs were obtained only for the baseline visit. In addition, when obtaining the quantiles, variable importance was estimated as the percentage in mean square error (%IncMSE), calculated by permuting sample values of the out-of-bag (OOB) in the test dataset, and increase in node purity (incNodepurity), calculated on the basis of the reduction in sum of squared errors for each decision tree; we rank-ordered the most important variable across all models in Appendix A) [29].

### 2.7. Predictive Performance

We estimated two CVD risk scores, (i) the Framingham risk score laboratory- and nonlaboratory-based [30], and (ii) the 2013 American College of Cardiology/American Heart Association Task Force [31]. Overall, both algorithms comprise data on age, sex, smoking, diabetes diagnosis, systolic blood pressure and its treatment, total cholesterol, and HDL-C. For the nonlaboratory-based risk model, BMI was used instead of lipids. We further compared the predictive ability (i.e., area under the receiver operating characteristic curve; ROC AUC) of two logistic regression models, one with the generated risk scores and one with risk score plus a variable indicating risk factor ‘sensitivity’ (Appendix A).

### 2.8. Time-to-Event Analysis

Cox proportional hazards regression models were used to estimate hazard ratios (HRs) and corresponding 95% confidence intervals (CIs) between sensitivity categories for each cardiometabolic trait derived from the QRF approach and the risk of diabetes and CVD-incidence and mortality. The proportional hazards assumption was tested with Schoenfeld residuals. The ‘neutral’ category was used as the reference group. Statistical significance (*p*-value) was set at the 5% level. Per cardiometabolic trait, a model including age and sex (and BMI, where this was not the outcome), fasting status, FFQ version, TEI, educational level (education was previously used as a proxy of socioeconomic status in this population [32]), smoking status, physical activity, and alcohol consumption. The covariates were selected a priori owing to their previously established associations with cardiovascular mortality in the Swedish population [33]; if a covariate was already in the environmental QRF model, it was not included. The timescale was the elapsed time from baseline in years until an event occurred or the study ended, whichever came first. HRs and 95% CIs were pooled for each cardiometabolic trait by sensitivity category to obtain an overall estimate under a random-effects model [34]; heterogeneity was assessed with Cochran’s Q statistic [35,36]. All statistical analyses were performed using *R* software version 3.6.1 [37]; statistical packages are listed in Appendix A.

## 3. Results

Baseline characteristics for each cohort are shown in Table 1. Median follow-up time (interquartile range (IQR)) for VHU was 9.7 (5.8) years and 21.1 (4.9) years for MDCS. In both cohorts, individuals classified as being ‘sensitive’ to lifestyle exposures affecting blood pressure and lipids had more cardiovascular events and deaths compared with the remainder of the population (all hazard ratios (HRs) and 95% CIs for CVD events, T2D, and CVD-mortality are in Appendix A).

### 3.1. Cardiovascular Events

In VHU, the risk of CVD in those who were classified as ‘sensitive’ to the lifestyle exposures affecting diastolic blood pressure was doubled, whereas, in MDCS, the risk in this same subgroup was increased by 32%, compared to the reference group. The risk of nonfatal and fatal CVD in people classified as sensitive to the lifestyle exposures affecting systolic blood pressure was ~60% and ~50% higher than the reference population for MDCS and VHU, respectively. When hazard estimates were pooled, the overall systolic and diastolic blood pressure ‘sensitive’ HRs were statistically significant under a random-effects model. In addition, the pooled groups of ‘sensitive’ individuals for systolic and diastolic blood pressure were also at higher risk for early death (Figure 2).

The risk of CVD in people classified as sensitive to LDL-C-related risk exposures was doubled in VHU and ~60% higher in MDCS, with the pooled estimate being statistically significant. In MDCS, those who were sensitive to lifestyle exposures lowering HDL-C were at higher risk of CVD, but this was not the case in VHU.

### 3.2. T2D Incidence

For glycemic traits, those classified as ‘sensitive’ in MDCS to the lifestyle risk factors for elevated fasting glucose had a fourfold increased risk of T2D. However, when risk estimates from MDCS were pooled with those from VHU, this result was not statistically significant (Table 2).

## 4. Discussion

Overall, a 50% to 60% higher risk of CVD and fatal CVD was observed in those individuals sensitive to the environments associated with blood pressure traits. Similarly, those with sensitivity to the environment related to LDL-C had 74% higher risk of CVD incidence. These findings are in line with others where higher blood pressure and dyslipidemia were shown to be associated with cardiovascular risk [39].

Public health guidelines to reduce disease risk rely on population-averaged estimates of risk factor susceptibility, often focusing on intermediate markers of cardiometabolic risk such as blood pressure or serum cholesterol levels. This strategy assumes that broad recommendations work well for most people, yet risk factor susceptibility and treatment response are highly heterogeneous [40], justifying public health interventions that are tailored to subgroups of the population. To explore whether doing so might be of clinical value, we used machine learning to identify, avoiding distributional assumptions, a population subgroup that is especially sensitive to modifiable lifestyle exposures for cardiometabolic disease. We showed that those who are especially sensitive to these risk exposures tended to develop CVD more rapidly. This type of risk classification is important, as it highlights individuals with ‘normal’ or ‘low’ levels of intermediate cardiometabolic markers, who are at relatively high risk of clinical events overlooked by conventional screening and risk classification approaches.

The approach we used focuses on sensitivity to modifiable risk factors trained on intermediate biomarkers of clinical disease. Not all of these intermediate marker sets proved informative. For example, sensitivity to obesogenic lifestyle factors did not raise the risk of T2D or CVD. Indeed, we found no clear evidence that sensitivity to lifestyle exposures in any biomarker set raised the risk of T2D. This may be because diagnosis of T2D is one of exclusion, where all known causes of chronically elevated blood glucose are eliminated, leaving the idiopathic label of T2D to be applied. Thus, T2D is highly heterogeneous in etiology and clinical presentation, making it harder to predict than more precisely defined diagnoses such as CVD. Nevertheless, as the wide confidence intervals around some of the risk estimates reported here indicate, it is likely that these analyses are underpowered, and some negative findings may be false positive.

Although these analyses benefited from comprehensive assessments of lifestyle exposures in these cohorts, a limitation is that they are predominantly self-reported data. Such data are prone to reporting biases, and some lifestyle factors are likely to have been assessed more precisely than others. Moreover, many variables prioritized from VHU were unavailable or captured differently in MDCS, which makes it difficult to isolate biological from statistical heterogeneity when pooled. The observational nature of the studies makes causal inference challenging, and one cannot rule out the possibility that some associations are confounded. There is little one can do to mitigate this common limitation of epidemiological studies. It might also be argued that to be classified as *sensitive* to adverse lifestyle exposures is a function of regression dilution, as this subgroup lies at the extreme of the prediction distributions, where measurement error will be greatest. However, this is unlikely in this setting, as sensitivity to lifestyle exposures persists across many years of follow-up. Nevertheless, trials are needed that assess whether people defined as *sensitive,* yet with apparently healthy biomarker profiles, are more susceptible to cardiovascular events than those who are not defined as *sensitive* and also benefit from intensive lifestyle interventions.

Most current clinical guidelines for T2D and CVD discuss the importance of personalized care, yet include generic lifestyle recommendations [41,42], overlooking between-person variability in susceptibility to environmental risk factors. There has been extensive debate about the role of precision medicine in disease prevention, which typically focuses on population subgroups with distinct risk factor and treatment response profiles, such that efficacy is maximized, and costs and risks are minimized [1]. The approach described here is aligned with the objectives of precision prevention, by identifying people at high risk of cardiometabolic disease and helping determine which modifiable exposures to intervene in. Strategies to prevent disease in this subpopulation may include nutritional support [43], lifestyle modification, and pharmacotherapy [44]; however, further investigation from randomized clinical trials is needed to discern which modality is more appropriate.

## 5. Conclusions

In conclusion, the approach to cardiometabolic risk stratification presented here may help improve the precision with which at-risk subgroups of the population are identified. In practice, the implementation of this approach would require combined assessments of modifiable risk exposures and intermediate markers of cardiometabolic risk. Calculating an individual’s level of risk using the current approach is more complicated than convention risk algorithms, because it leverages conditional probabilities. However, this could be managed through app-based assessment and decision support systems, which have proven successful elsewhere [45].

## Figures and Tables

**Figure 1 nutrients-14-03171-f001:**
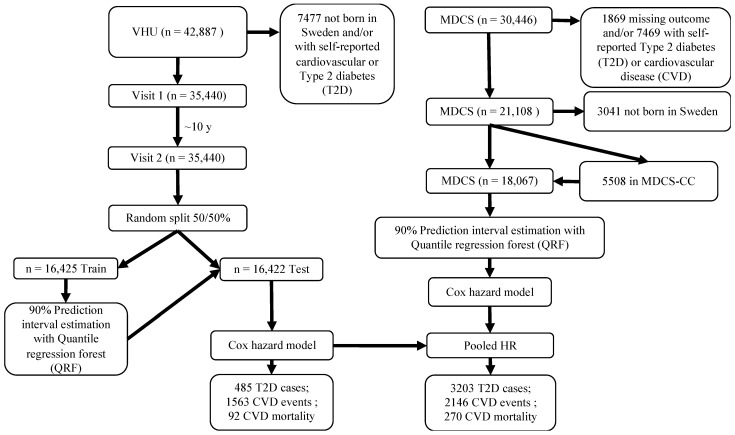
Study flowchart of VHU and MDC studies, data processing, and model training. VHU: Västerbotten Health Survey; MDCS: Malmö Diet and Cancer Study; MDCS-CC: MDCS Cardiovascular Cohort.

**Figure 2 nutrients-14-03171-f002:**
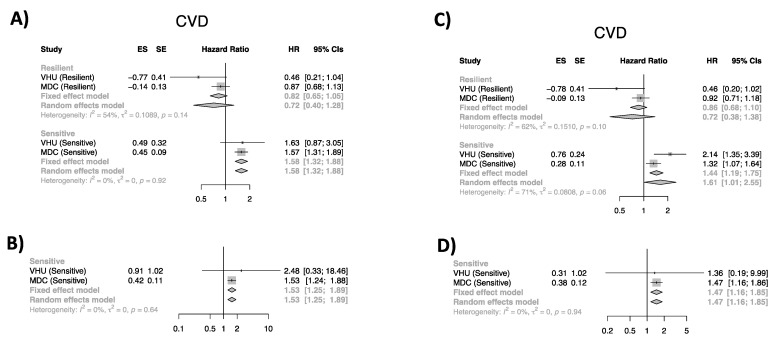
Forest plots of pooled studies by persistence category and CVD event. (**A**,**C**) Systolic blood pressure (SBP (mm/Hg)); (**B**,**D**) diastolic blood pressure (DBP (mm/Hg)). Random- and fixed-effects meta-analysis of the association between trait-persistence category and CVD and CVD mortality. For (**C**,**D**), the number of events did not allow to obtain pooled estimates for the ‘resilient’ group. The square and diamond shapes represent summary estimates, while the horizontal bars represent the 95% confidence intervals. HR: hazard ratio; ES: effect estimate; SE: standard error; CVD: cardiovascular disease.

**Table 1 nutrients-14-03171-t001:** Baseline characteristics of study cohorts.

	VHU	MDCS
n	35,440	18,067
Male (%)	15,599 (46.8)	6772 (37.5)
Age	42.96 (7.02)	57.72 (7.71)
BMI (kg/m^2^)	25.10 (3.71)	25.30 (3.78)
Total cholesterol (mmol/L)	5.47 (1.14)	6.20 (1.11)
HDL-C (mmol/L)	1.32 (0.57)	1.40 (0.37)
LDL-C (mmol/L)	3.92 (1.16)	4.19 (1.02)
Triglycerides (mmol/L)	1.32 (0.76)	1.47 (0.75)
Fasting glucose (mmol/L)	5.31 (0.63)	5.02 (0.83)
2 h glucose (mmol/L)	6.39 (1.30)	-
HbA1c (mmol/mol) ^a^	-	31.4 (5.05)
Systolic blood pressure (mm Hg)	123.27 (15.77)	138.58 (18.97)
Diastolic blood pressure (mm Hg)	77.25 (10.86)	84.02 (9.53)

All values are the mean (SD) unless otherwise stated. VHU: Västerbotten intervention program; MDCS: Malmö Diet and Cancer; BMI: body mass index; HDL-C: high-density lipoprotein cholesterol; LDL-C: low-density lipoprotein cholesterol; HbA1c: glycated hemoglobin; 2 h glucose: 2 h glucose tolerance. ^a^ Raw value collected in DCCT (Diabetes Control and Complications Trial) units, transformed to mmol/mol units using formula HbA1c (mmol/mol) = 10.929 × (HbA1c (%) − 2.15) [38]. *Note*: To convert to mg/dL multiply cholesterol by 38.67, blood glucose by 18.0182, and triglycerides by 38.67.

**Table 2 nutrients-14-03171-t002:** Pooled hazard ratios (HR) and 95% CI and outcomes from VHU and MDCS.

	CVD			Test between Groups ª		T2D			Test between Groups ª		CVD Mortality			Test between Groups ª	
Trait	HR	95% (CIs)		Q	*p*	HR	95% (CIs)		Q	*p*	HR	95% (CIs)		Q	*p*
Fasting glucose															
Pooled neutrality	1.00					1.00					1.00				
Pooled resilient	0.77	0.31	1.90	0.25	0.62	0.73	0.46	1.16	0.75	0.39	1.04	0.61	1.75	0.12	0.73
Pooled sensitive	1.01	0.55	1.86			1.69	0.26	10.87			1.18	0.69	2.03		
^b^ 2 h Glucose/HbA1c															
Pooled neutrality	1.00					1.00					1.00				
Pooled resilient	0.77	0.54	1.12	5.41	0.02	0.62	0.08	4.55	0.36	0.55	0.75	0.39	1.47	0.74	0.39
Pooled sensitive	1.46	0.99	2.17			1.23	0.46	3.31			1.11	0.62	2.00		
Diastolic blood pressure															
Pooled neutrality	1.00					1.00					1.00				
Pooled resilient	0.72	0.38	1.38	3.88	0.05	0.64	0.26	1.55	0.15	0.70	1.05	0.81	1.37	3.45	0.06
Pooled sensitive	1.61	1.01	2.55			0.81	0.36	1.82			1.47	1.16	1.85		
HDL-C															
Pooled neutrality	1.00					1.00					1.00				
Pooled resilient	1.21	0.50	2.98	0.03	0.87	2.22	0.96	5.12	1.12	0.29	1.39	0.79	2.44	0.01	0.94
Pooled sensitive	1.12	0.67	1.84			0.69	0.10	5.03			1.47	0.38	5.62		
BMI															
Pooled neutrality	1.00					1.00					1.00				
Pooled resilient	1.07	0.84	1.37	0.59	0.44	1.37	0.30	6.24	0.98	0.32	1.57	1.20	2.06	1.70	0.19
Pooled sensitive	0.86	0.51	1.44			0.59	0.31	1.13			1.22	0.93	1.60		
LDL-C															
Pooled neutrality	1.00					1.00					1.00				
Pooled resilient	1.34	0.91	1.98	0.99	0.32	0.59	0.24	1.44	0.03	0.87	1.31	0.80	2.15	0.21	0.65
Pooled sensitive	1.75	1.24	2.46			0.65	0.29	1.48			1.72	0.60	4.97		
Total Cholesterol															
Pooled neutrality	1.00					1.00					1.00				
Pooled resilient	1.17	0.55	2.51	0.32	0.57	1.07	0.62	1.85	0.20	0.66	1.58	0.99	2.53	0.23	0.63
Pooled sensitive	1.58	0.78	3.19			1.30	0.67	2.53			1.25	0.53	2.92		
Triglycerides															
Pooled neutrality	1.00					1.00					1.00				
Pooled resilient	1.09	0.66	1.78	0.01	0.94	-	-	-	-	-	0.84	0.44	1.59	1.53	0.22
Pooled sensitive	1.06	0.74	1.52			1.04	0.48	2.25			1.39	0.85	2.29		
Systolic blood pressure															
Pooled neutrality	1.00					1.00					1.00				
Pooled resilient	0.72	0.40	1.28	6.55	0.01	0.74	0.38	1.47	3.17	0.07	1.01	0.77	1.32	5.74	0.02
Pooled sensitive	1.58	1.32	1.88			1.65	0.95	2.84			1.53	1.25	1.89		

ª Test for subgroup differences between resilient and sensitive groups; ^b^ VHU; ‘-’ indicates that it was not possible to estimate the number. Pooled estimates were obtained with inverse variance method and DerSimonian–Laird estimator for random-effects models; HDL-C: high-density lipoprotein cholesterol; LDL-C: low-density lipoprotein cholesterol; BMI: body mass index; HbA1c: glycated hemoglobin CVD: cardiovascular disease. T2D: type 2 diabetes. Adjustment for each cohort model included age, sex, BMI, fasting status, FFQ version, TEI, educational level, smoking status, physical activity, and alcohol intake.

## Data Availability

The individual-level data from VHU and MDCS are not publicly available due to privacy and consenting constraints. However, applications for data access can be submitted to the Department of Biobank Research, Umeå University (https://www.umu.se/en/biobank-research-unit/; accessed on 14 February 2020) and Lund university (https://www.malmo-kohorter.lu.se/malmo-cohorts; accessed on 20 September 2021) for the VHU and MDCS cohorts, respectively.

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
