# Peer review of "Predicting Sensitivity to Adverse Lifestyle Risk Factors for Cardiometabolic Morbidity and Mortality"

_nutrients, 2022, doi:10.3390/nu14153171_

Round 1
Reviewer 1 Report
Dear authors and editor,
The manuscript titled "Predicting sensitivity to adverse lifestyle risk factors for cardio-metabolic morbidity and mortality" This is a cohort study that ascertain a person’s degree of resilience or sensitivity to adverse lifestyle exposures and determine whether these classifications help predict cardiometa-bolic disease later in life.
There are many minor and major issues I'd like the authors resolve.
Abstract
1-Add the study design to the abstract or title.
2- Change the keywords. Delete the words "cardiometabolic disease"; "quantile random forests"; "prediction interval" and "risk sensitivity". Not found in the MeSH (Medical Subject Headings). Change to "Cardiometabolic Risk Factors" "Risk Assessment" ......
3- It is recommended not to include abbreviations in the abstract or abbreviations that have not been previously explained.
Introduction
4-It is recommended that this section be expanded.
Materials and Methods
5-Study size: Explain how the study size was arrived at. There are clear differences in the number of participants between the two cohorts (35,440 vs. 18,067).
I am concerned about the differences in data collection between the two cohorts. Explain the reasons for these differences.
"In VHU, physical activity was assessed using the modified version of the International Physical Activity Questionnaire [16,17], and a validated semi-quantitative food frequency questionnaire (FFQ), designed to capture habitual diet over the last year, was used to obtain information on various dietary factors [18]. In 1996, the FFQ was reduced from 84 to 66 items by merging similar items and removing those deemed redundant. For MDCS, a modified diet history method consistingof a 7-day food diary covering all cooked meals, a 168-item FFQ covering the non-cooked meals for the previous year were administered. Moreover, a 1-hour interview was used to determine portion sizes, cooking methods and food choices. Nutrient and energy contents were calculated using the Swedish Food Composition Database"
On the other hand, the rest of the methodology meets the quality standards. The design, the study variables, the ethical permission and the statistical analysis are specified.
Results
- adequate
Discussion
- adequate
The authors point out the limitations related to the design used and the sample collection.
6-It is recommended to start the discussion by indicating the most relevant findings in the results.
Conclusion
- adequate
Reference:
- adequate
Reviewer 2 Report
The topic of the manuscript is interesting but should be better explained. Indeed, identifying and/or preventing cardiometabolic diseases is a great goal. I have some suggestions to improve the paper:
1. The English form is poor. Please, revise the manuscript with an English native speaker.
2. The quality of the Tables should be increased (particularly Table 1).
3. There is a lacking iconography (just a flow-chart).
4. The introduction should better focus on the effects of cardiometabolic diseases on patients. Please, find some papers to cite and discuss:
- Eur J Intern Med 2022 May;99:89-92. doi: 10.1016/j.ejim.2022.03.012. Epub 2022 Mar 14. Physical Decline and Cognitive Impairment in Frail Hypertensive Elders During COVID-19.35300886
- Cardiovasc Diabetol 2022 Jan 19;21(1):10. doi: 10.1186/s12933-021-01442-z. Correlation of Physical and Cognitive Impairment in Diabetic and Hypertensive Frail Older Adults. 35045834
- Nutrients 2019 Jan 4;11(1):85. doi: 10.3390/nu11010085. The Role of Nutrients in Reducing the Risk for Noncommunicable Diseases During Aging. 30621135
- Nutrients 2017 Aug 9;9(8):848. doi: 10.3390/nu9080848. Cardio-metabolic Benefits of Plant-Based Diets. 28792455
- Nutrients 2020 Jun 30;12(7):1955. doi: 10.3390/nu12071955. Ultra-Processed Foods and Health and Outcomes: a Narrative Review. 32630022
5. The discussion should speculate on potential treatments and future perspectives. Some papers are focusing on pleiotropic effects of SGLT2 inhibitors. Please, cite and discuss:
. Eur Heart J. 2022 Mar 14;43(11):1029-1030. doi: 10.1093/eurheartj/ehab765. SGLT2 inhibitors: the statins of the 21st century. 34741610
- Hypertension 2022 Jun 15;101161HYPERTENSIONAHA12219586.doi: 10.1161/HYPERTENSIONAHA.122.19586. SGLT2 Inhibition via Empagliflozin Improves Endothelial Function and Reduces Mitochondrial Oxidative Stress: Insights From Frail Hypertensive and Diabetic Patients. 35703100
- Diabetes Care 2022 May 1;45(5):1247-1251. doi: 10.2337/dc21-2434. Empagliflozin Improves Cognitive Impairment in Frail Older Adults With Type 2 Diabetes and Heart Failure With Preserved Ejection Fraction 35287171
- Lancet Diabetes Endocrinology. 2021 Sep;9(9):586-594. Dapagliflozin in patients with cardiometabolic risk factors hospitalised with COVID-19 (DARE-19): a randomised, double-blind, placebo-controlled, phase 3 trial. 34302745
- Diabetes Obes Metab. 2020 Dec;22(12):2384-2397. doi: 10.1111/dom.14164. Epub 2020 Sep 10. Cardiometabolic risk factor control in black and white people in the United States initiating sodium-glucose co-transporter-2 inhibitors: A real-world study. 32744394.
Round 2
Reviewer 1 Report
I am satisfied with the authors' response. However, I recommend that they include in the limitations the possible biases between the two cohorts due to differences in sample size and the use of different tools when collecting the variables.
The authors themselves comment on this.
"We agree with the reviewer that differences in: (a)sample size, and (b)tools to capture environmental variables vary and may impact the estimates when pooled. "
While it is true that there are no significant differences, the reader should bear this in mind, especially when dealing with subgroups.
Author Response
We thank the Reviewer for her/his comments. We have now added the following limitation in the Discussion section:
“Although these analyses benefited from comprehensive assessments of lifestyle exposures in these cohorts, a limitation is that they are predominantly self-reported data. Such data are prone to reporting biases and some lifestyle factors are likely to have been assessed more precisely than others. Moreover, many variables prioritised from VHU were unavailable or captured differently in MDCS, which makes it difficult to isolate biological from statistical heterogeneity when pooled...”
Reviewer 2 Report
The authors have improved the manuscript. In particular, the iconography is surely improved.
Author Response
We thank Reviewer 2 comments.